# State of the Art of Monitoring Technologies and Data Processing for Precision Viticulture

**Marco Ammoniaci** [ID], **Simon-Paolo Kartsiotis** *[ID], **Rita Perria and Paolo Storchi** [ID]

CREA—Council for Agricultural Research and Economics, Research Centre for Viticulture and Enology, Viale Santa Margherita, 80, 52100 Arezzo, Italy; marco.ammoniaci@crea.gov.it (M.A.); rita.perria@crea.gov.it (R.P.); paolo.storchi@crea.gov.it (P.S.)

\* Correspondence: simonpaolo.kartsiotis@crea.gov.it; Tel.: +39-3341611876

**Abstract:** Precision viticulture (PV) aims to optimize vineyard management, reducing the use of resources, the environmental impact and maximizing the yield and quality of the production. New technologies as UAVs, satellites, proximal sensors and variable rate machines (VRT) are being developed and used more and more frequently in recent years thanks also to informatics systems able to read, analyze and process a huge number of data in order to give the winegrowers a decision support system (DSS) for making better decisions at the right place and time. This review presents a brief state of the art of precision viticulture technologies, focusing on monitoring tools, i.e., remote/proximal sensing, variable rate machines, robotics, DSS and the wireless sensor network.

**Keywords:** DSS; WSN; remote sensing; proximal sensing; variable-rate technology; grapevine; vineyard; UAV



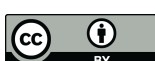

## 1. Introduction

Precision agriculture (PA) is defined as an agricultural, forestry and livestock management based on the observation, measurement and response of the set of inter and intra-field quantitative and qualitative variables that act in agricultural productions. This is in order to define a decision support system (DSS) for the entire farm management, with the aim of optimizing yields looking at climate, environmental, economic, productive and social sustainability. The synthesis of this definition was effectively given by Pierce and Novak [1], who identified PA with "doing the right thing, in the right place at the right time". The starting point of the PA process is the collection of data, which can be done through proximal or remote sensors. Then, these collected data are interpreted and evaluated by an agronomic point of view in order to traduce them into manual implementations or into inputs for variable rate technology (VRT) machines, which are able to perform the prescribed actions in a semi-automatic or fully automatic way.

The general working scheme of PA is represented in Figure 1 [2].

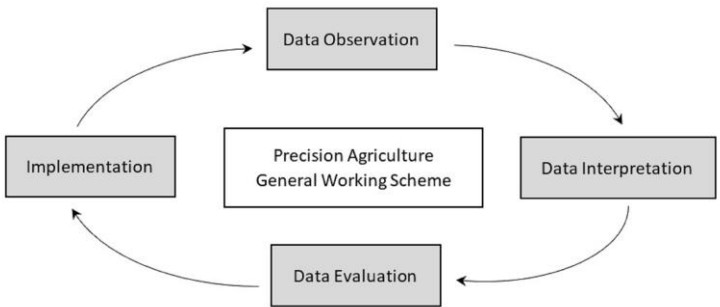

**Figure 1.** Precision agriculture (PA) general workflow [2].

The hypothesis behind PA is that each field is not uniform, as it is generally considered in conventional agriculture, but it is managed taking into account its site-specific needs. This management strategy increases the efficiency of agricultural inputs and, if correctly adopted, it results in cost savings and increased benefits [3,4]. Furthermore, the site-specific dosing of inputs in a field allows for an optimized use of resources from an environmental and food safety point of view [5], since overdose and therefore loss of nutrients, pesticides and water are prevented, especially where plants do not have real deficiencies or stress [5].

According to several previous studies, in general, the application of PA positively increases profits, as it has been demonstrated in different crops and for various adopted technologies [6,7].

The proven benefits of PA are many and can be summed up as following:

- Maximization of yields;
- Identification of plant stress;
- Constant monitoring of crops with the possibility to implement targeted actions;
- Reduction of intra-field variability;
- Reduction of costs and time of agricultural operations;
- Lower the environmental impact of agricultural operations;
- Optimization of the use of fertilizers, pesticides and water;
- Increase of products quality.

In Table 1, the main PA parameters collected by different monitoring systems have been reported along with their benefits.

**Table 1.** PA monitoring systems, data collection and benefits.

| Type of Monitoring | Collected Data | Benefits |
|---|---|---|
| Meteorological | Temperature<br>Air humidity<br>Leaf wetness<br>Wind speed<br>Rainfall data<br>Solar radiation | Disease prevention, pesticide and water management |
| Soil | Electric conductivity<br>Soil Texture<br>Organic matter content<br>pH<br>Humidity | Fertilization, seed and water management, project of new plantations |
| Plant | Vigor and biomass<br>LAI (leaf area index)<br>Fluorescence<br>Growth ratio | Fertilization, pesticide, harvest, defoliation and water management |
| Fruit | Ripening grade<br>Sugars<br>Anthocyanin<br>Acidity<br>Growth ratio | Selective harvesting to increase the quality of the final products (e.g., wine) |

The adoption of PA implies to adapt agronomic inputs like fertilizers, pesticides and water according to the specific needs of each area of the field [8].

The practical implementation of PA can be done using various technologies: crop and yield sensors, proximal and remote sensors, GNSS (global navigation satellite system) sensors, VRA (variable rate application) equipment and VRT machines, GIS (geographic information systems) systems for data analysis and interpretation [9,10].

The real strength of PA lies in the use of all these technologies in combination with each other, so that each phase of cultivation is monitored in order to make targeted and effective decisions.

An overview of PA technologies is reported in Figure 2 [11].

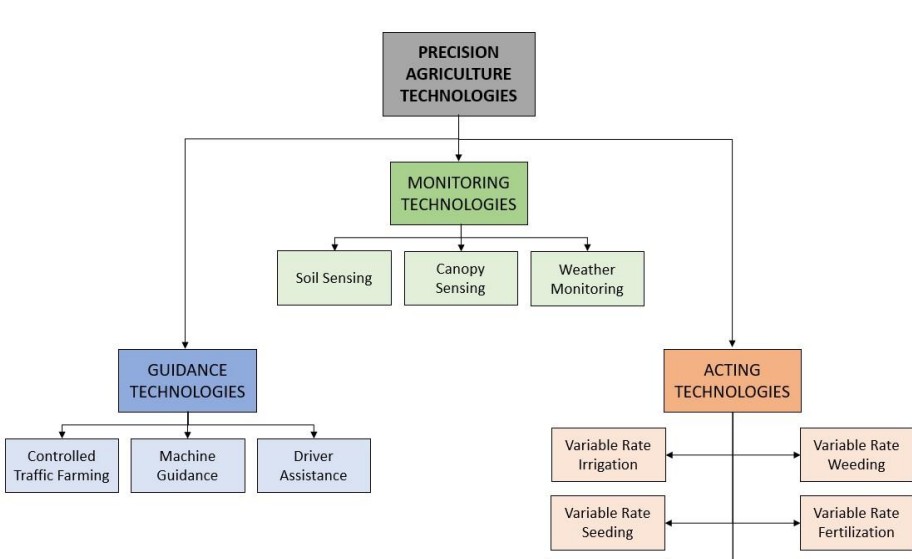

**Figure 2.** Precision agriculture technologies overview (adapted from [11], with permission from *Sustainability*, 2017).

One of the sectors most sensitive to the application of PA technologies is viticulture. In fact, in viticulture, it is common to find that within the same vineyard there are areas characterized by different soil composition and structure [12,13], by different concentration of humidity, by different solar exposure and microclimate: to these differences the vine responds accordingly, highlighting different physiological expressions. Vegetative vigor is the most obvious of these responses, which can be assessed by vegetation indices like the NDVI (normalized difference vegetation index).

The main goal of precision viticulture (PV) is to assess health, vigor and physiological needs of the vines belonging to different areas of the vineyard in order to adapt the standard cultivation techniques in a more site-specific and time-specific manner. To achieve this, winegrowers have to use IT tools able to manage a huge number of data in a totally or almost totally automated way [14]. The collection of georeferenced spatial datasets provides winegrowers with the opportunity to optimize the decision-making process by using several management techniques used to minimize the yield variability but also to take advantage of this variability for improving grape/wine quality [10], while saving on management and input costs [6,7,15]. About economic savings, many researchers have demonstrated that the implementation of PV can increase grapes quality and yield along with savings in production costs [16,17].

Research areas of PV have focused on four main specific fields [10,18]:

1. quantification and evaluation of intra-field variability;
2. delimitation of differential treatment areas at parcel level, based on the analysis and interpretation of this variability;
3. development of variable rate technologies (VRT);
4. evaluation of opportunities for site-specific vineyard management.

Research in these fields allows winemakers, oenologists and agronomists to know and understand the variability within the same field, which are the causes of this variability, how yield and quality are correlated with it and if the site-specific management of the vineyard is justifiable on a technical and economic basis.

## 2. Story

PA is a relatively new discipline that was developed in a more scientific way only in the mid-1980s and it has been listed in the top ten agricultural developments in recent decades [19].

PA has spread out mainly thanks to four factors [19]:

1.  the availability of accurate and cheap global navigation satellite systems (GNSS);
2.  the development of GIS software to visualize/analyze spatial and geographical data;
3.  the growing availability of georeferenced information acquired remotely;
4.  the development of variable rate technologies (VRT).

The first known and recorded attempt of implementing PA was in the 1920s when Linsley and Bauer sampled the soil pH to create a prescription map of a corrective dose (limestone) which was then distributed by hand [20]. In the 1960s, geo-statistics contributed considerably in providing tools to manage space-time variability [21]. In the 1980s, the first sensors to measure soil geo-electric properties came on the market [15]. During the 1990s, GPS [15] for civilian use came available, allowing for GNSS guidance in tractors. Additionally, the first VRT fertilizer was used and satellite and aerial images were used to discriminate areas with different characteristics in the field. During this period, optical sensors mounted on farm machinery were starting to be used to monitor the vegetative vigor [15]. At the end of the century, the ISOBUS standard, a communication protocol between tractors and operating machines, was developed. In the last 20 years, the new big protagonists are satellites and UAVs (unmanned aerial vehicles or drones), which have made it possible to have low-cost images at a very high resolution.

In Table 2, PA diffusion in the world by the technique adopted has been summarized.

**Table 2.** PA world diffusion among farmers by technique [15,22].

| Country | PA Technique | Diffusion |
|---|---|---|
| USA | Tractors with GPS | 80% |
| | Autosteer tractors | 40% |
| | VRT fertilization | 70% |
| | VRT seeding | 60% |
| | VRT spraying | 38% |
| | Soil sampling | 70% |
| | Soil electrical conductivity mapping | 25% |
| | Harvester with production mapping system | 40% |
| | Yield monitoring | 58% |
| | UAV mapping | 38% |
| | Satellite mapping | 55% |
| UK | Tractors with GPS | 22% |
| | Soil mapping | 20% |
| | Variable rate applications | 16% |
| | Production mapping | 11% |
| Germany | Various PA techniques | 10% |
| | ISOBUS systems | 45% |
| France | Various PA techniques | 10% |
| | Variable rate applications | 10% |
| | ISOBUS systems | 30% |
| Italy | Tractors with GPS | 8% |
| | ISOBUS systems | 10% |
| | VRT fertilizer | 200 machines |
| | Harvester with production mapping system | 1600 machines |

The implementation of PV techniques is relatively recent [18] and occurred much later than other crops and only after the mid-2000s. This delayed start is due not to the lack of interest by winegrowers, but to difficulties intrinsically associated with the characteristics

of the vineyard, such as a canopy with a discontinuous character and an organization in rows, which requires a very high resolution of the images to discriminate the canopy from the soil and a big data processing capacity to manage the spatial information before use [9,23].

The most relevant aspects that must be taken into account in the PV include the optimization of inputs, differential grape harvesting to produce higher quality wines [24], yield forecasting and greater accuracy and efficiency of canopy/soil sampling conducted at parcel level [2,25,26].

The main objective of PV, in general, coincides with the main objectives of PA: the appropriate management of the variability of crops, an increase in economic savings and a reduction of the environmental impact [27,28].

## 3. Monitoring Systems

### 3.1. Remote Sensing

The remote sensing systems are characterized by optical sensors like visible cameras (RGB), multispectral sensors, that detect specific bands for the analysis of vegetation, thermal sensors, useful for measuring the temperature of plants and identifying water stress, and hyperspectral sensors, which allow to carry out in-depth analyzes on crops and for the study of diseases.

Remote sensing technologies, like satellites and UAVs, have been widely used to assess management zones in vineyards in order to increase the wine quality [29,30] and to estimate berry phenolics and color at harvest [31].

In Table 3, the comparison between different remote sensing platforms is reported in terms of time/spatial capabilities and from an economic point of view [22].

**Table 3.** Remote sensing platforms comparison [22].

| Platform | Spatial Resolution | Autonomy | Coverage Area | Cost per Hectare |
|---|---|---|---|---|
| Satellite | 0.4–100 m | Unlimited | Thousands of ha | Free–0.3 €/ha |
| Aircraft | 10–100 cm | 1–3 h | Hundreds of ha | 100–500 €/ha |
| UAV | 0.5–10 cm | 30–60 min | Tens of ha | 60–120 €/ha |

### 3.1.1. Satellites

Satellites have been used in PA for over 40 years, when Landsat 1 was launched in 1972. It was equipped with a multispectral sensor and provided a spatial resolution of 80 m with a revisit time of approximately 18 days. The first application in PA of Landsat images was to estimate the spatial distribution for soil organic matter content [32,33].

The last launched Landsat mission satellite is the Landsat 8 [34,35], which operates in the visible, near-infrared, short wave infrared and thermal infrared spectrums.

The use of satellites in remote sensing has great potential, but spatial resolutions are not sufficient for PV due to the presence of the inter-rows and the width of the rows themselves, which are less than one meter thick. Image processing in order to remove the soil and inter-row pixels is a rather complicated operation and intrinsically not possible with image resolutions greater than 0.5 m. Another limitation is due to the temporal resolution and cloud cover that can occur at the time of satellites data capture.

One of the most used satellites in PA and PV is the Sentinel-2 one [36]. It carries an optical instrument called MSI (multi-spectral instrument) capable of sampling 13 spectral bands down to a resolution of 10 m. The main advantage of the Sentinel-2 with respect to other satellites is that the data are open-source, so totally free of charge and freely downloadable from various websites.

Another high-resolution satellite is the RapidEye [37], which acquires images in 5 multispectral bands (blue, green, red, red edge, near infrared) at a resolution of 5 m. RapidEye has been used to evaluate NDVI in order to characterize the vine vigor and some

technological and phenolic parameters. Simple linear relationships between NDVI at berry set, pre-veraison and ripening has been found to evaluate sugar content and anthocyanins at harvesting [38]. RapidEye has also been used to evaluate the LAI in "tendone" vineyards demonstrating a good correlation with in-field estimation [39].

Among the very high-resolution satellites, the WorldView [40] mission is made up of 4 satellites: WorldView-1, WorldView-2, WorldView-3 [41] and WorldView-4 (dismissed due to failure). WorldView-3, launched in 2014, is the first commercial ultra-high-resolution satellite with 29 spectral bands and an average revisit time of less than 1 day.

For heatwaves in grapevines, WorldView imagery has been compared with Sentinel-2, showing a good agreement between the two datasets in discriminating both vigor and heat stress [42]. In addition, WorldView has been demonstrated to detect vineyards, extract vine canopy and discriminate several varieties [43].

An overview of main satellites used in PA and PV is reported in Table 4.

**Table 4.** Satellites most used in PA and PV with acquired bands, resolutions and applications.

| Satellite | Bands (Spatial Resolution) | Applications |
|---|---|---|
| Landsat 8 | 1 panchromatic band (15 m)<br>8 multispectral bands (30 m)<br>2 thermal infrared bands (100 m) | Agriculture and forestry, environmental monitoring, geology, land use mapping, hydrology, coastal resources [35] |
| Sentinel-2 | 13 multispectral bands (10 m/20 m/60 m) | Land monitoring, maritime monitoring, emergency management, security [36] |
| RapidEye | 5 multispectral bands (5 m) | Agriculture and forestry, environment, mapping, defense, security and emergency, visual simulation [37] |
| WorldView 3 | 1 panchromatic band (0.31 m)<br>8 multispectral bands (1.24 m)<br>8 shortwave infrared bands (3.7 m)<br>12 CAVIS (Clouds, Aerosols, Vapors, Ice, and Snow) bands (30 m) | Mapping, land classifications, disaster preparedness/response, feature extraction, soil/vegetative analysis, geology, environmental monitoring, bathymetry, coastal applications [41] |

### 3.1.2. Aircrafts

Aircrafts allow to monitor large areas with a long flight range and also by carrying heavy and big payloads, thus providing the ability to manage multiple sensors at one time. Aircrafts bypass some limitations of satellites by scheduling image time capture and providing higher ground resolution (down to 10 cm), depending on the flight altitude. However, the reduced flexibility of time acquisition, due to the rigid flight schedule, and high operating costs, makes it economically feasible only on areas bigger than 10 ha [9].

Additionally, as for UAVs, aircrafts undergo the National Civil Aviation Authority in force, so that the pilot has to be licensed and authorized to fly over the survey zone. In fact, in each country, there are zones not flyable like prisons, military buildings or areas, airports, sensitive targets, restricted, prohibited and dangerous areas. Each Civil Aviation Authority publishes detailed maps and lists about these areas in order to inform pilots if in the targeted survey zone there are some aerial activities (e.g., military exercises) and in which flight altitudes range, hours, days and months they can take place.

### 3.1.3. UAV

With the increasing competitiveness in the wine market, along with the increasing need for sustainable use of resources, the optimization of farm management is becoming essential. Thanks to photogrammetric techniques and to technological development in the field of automation, images acquired from unmanned aerial vehicles (UAVs or drones, Figure 3) can provide valuable information to winegrowers, giving a support to decision-making processes [44].

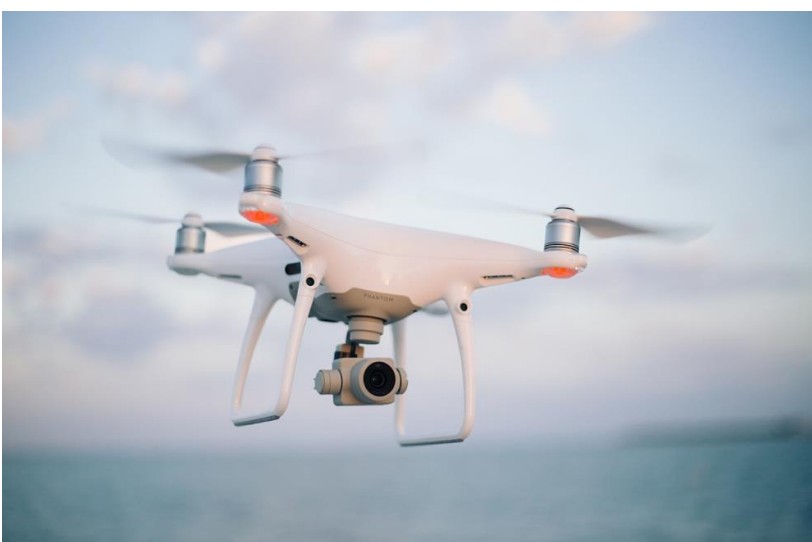

**Figure 3.** Rotary-wings UAV.

These fixed or rotary wings platforms are able to fly autonomously thanks to flight software or apps and can be controlled remotely by a ground pilot. UAVs embed flight control sensors (gyroscopes, compass, GPS, pressure sensor and accelerometers) controlled by a microprocessor. These platforms can be equipped with a variety of sensors, allowing a wide range of monitoring tasks to be performed. UAVs enable a very high spatial resolution on the ground (down to cm) and the possibility of highly flexible and timely monitoring, thanks to the reduced planning times. These characteristics make them ideal for small-to-medium fields (1–20 ha), especially in areas characterized by high fragmentation due to high heterogeneity.

The UAV mission planning is quite easy and can be done by using mobile apps (e.g., Drone Deploy, Pix4D Capture, Map Pilot) and open source software (Mission Planner): the area to be mapped is simply drawn by hand and the flight parameters are set up (flight altitude, flight speed, front/side overlap of the images to be captured). A grid mission (i.e., serpentine path) is set up to map agricultural areas. When using these apps/software in hilly environments, one feature to be used is the terrain following, i.e., the capability to follow the terrain slope when flying the UAV: this is crucial to maintain a fixed altitude with respect to the ground throughout all the mission, thus allowing a fixed GSD (ground sampling distance) which is fundamental to get a high quality map.

Depending on the application scenario (flat land, hill, field extension), it is possible to choose fixed-wings or rotary-wings UAVs. In particular, fixed-wing UAVs allow to monitor large areas (tens of hectares) more effectively, especially if not fragmented and in flat areas, having a high operational speed and high flight times (up to 1 h), but require more piloting skills and sufficient landing spaces. On the other hand, rotary-wings UAVs are easier to fly and allow better monitoring for small areas (up to 20 hectares), even fragmented and also with some orography, having high operational versatility and low flight times (up to 30 min).

With respect to satellites, UAVs have a much higher resolution, thus some considerations have to be done when assessing for vigor or water stress in tree crops as vineyards. Generally, the use of UAVs with high-resolution sensors instead of satellites is suggested due to the tree structure and inter-rows spacing of vineyards, which significantly affect the capacity to detect variability by sensors. Satellites images with different resolutions have been demonstrated to show similar behavior in assessing vineyards variability but not as good as UAV, unless high-resolution satellites are used [45,46]. Additionally, as it has been demonstrated in [47], bare soil has a negative effect on assessing vineyard vigor, while weeds or grassing tend to increase the actual vigor. In this way, a pixel containing vigorous plants plus bare ground would describe a less than real situation and incorrect vigor maps

would be generated. UAVs can provide images with a high detail that discriminates the row from the inter-row, thus providing pure pixels of only canopy, as opposed to satellites in which the pixels of higher dimensions incorporate both canopy and soil or grassing, thus altering the actual plant vigor or temperature [47].

Despite the positive aspects highlighted, the UAV platforms have two important limitations: the first is the autonomy, which is usually less than 30 min for rotary-wings UAVs and 60 min for fixed-wings UAVs, while the second is operational. In fact, in order to fly safely according to the National Civil Aviation Authority in force, the UAV pilot is required to have a suitable flight license and civil liability insurance in case of accidents and damage to third parties. The UAV must also be registered on appropriate platforms (each country has its own one), which issues a sort of "license plate" that must be applied to the UAV in order to make it recognizable.

Lightweight commercial UAVs with RGB, multispectral and thermal sensors are already available at prices lower than 10,000 €. These UAVs usually weigh less than 5 kg and are equipped with a sensor sensitive to sunlight, positioned on the upper part of the body, which captures the solar radiation in order to maximize the accuracy of the data collected over different times of day and year. Integrated GPS RTK (real time kinematics) modules are often present to provide accurate, real-time positioning data for each image, optimizing photogrammetric results and offering accurate measurements down to the centimeter. Furthermore, pressure sensors and obstacle-avoid sensors are also present to guarantee a safe flight. The resolution of the images acquired by UAVs can be down to 0.5 cm/pixel, allowing for a very detailed analysis on vegetation and soil.

Due to these characteristics, UAVs with RGB and multispectral or thermal sensors have been used extensively in viticulture to detect vineyards and vine rows [48], estimate grape yield [49], assessing for vegetative vigor [47] and water stress [50,51], detect missing plants [47], assess grapes maturity [51]. Limited to trellis system vineyards, a comparison between UAV and Sentinel-2 NDVI maps to assess vigor spatial variability has shown the same trend [52].

From the economic point of view, UAV technology has been assessed by various research studies as compatible with the agronomic costs, especially in vineyards with more than 40 hectares or for agricultural cooperatives with lots of fragmented vineyards [14,53–55].

### 3.2. Proximal Sensing

The knowledge of the spatial and temporal variability of the characteristics of vegetation and soil has important implications in many areas of production activities and environmental monitoring. The need to monitor changes in the state of the crops over time and space has led to the development of alternative sensing techniques with respect to conventional destructive and invasive methods, which also provide limited spatial coverage. Through these techniques, it is possible to obtain information quickly and relatively at low cost regarding soil, vegetation cover, nutritional status, efficiency of the photosynthetic system and the evapotranspiration process, water status, concentration of pigments, phytosanitary status and production response [15]. However, proximal sensors require site-specific calibrations and, in some cases, the data analysis can be complex.

Proximal sensors are generally of two types: optical or by contact. In Table 5, an overview of proximal sensing sensors and application is reported.

**Table 5.** Proximal sensing sensors and relative applications.

| Type of Sensor | Applications |
| --- | --- |
| Radiometric Fluorometer Apps (VitiCanopy) | Canopy vigor/stress assessment, chlorophyll content, nitrogen concentration, LAI, water stress |
| Geophysical Spectroradiometers | Soil composition and structure |
| Fluorometer Spectrophotometer | Grape quality and ripening assessment |

3.2.1. Canopy

Radiometric Sensors

The use of proximal sensors operating in the optical domain is based on the measurement of the electromagnetic radiation reflected or emitted by bodies and it can allow the rapid collection, in a non-destructive way and on large surfaces, of important information for the indirect measurement of the vegetation state.

The spectral response of vegetation is characterized by some characteristic features, mainly related to the radiation absorption by the pigments, which allows the identification of the presence of vegetation cover but also the characterization of its health state [15]. These reflectance characteristics are the basis for the use of proximal sensors for crops monitoring and management. In fact, particularly relevant is the strong contrast in reflectance between red and near infrared bands and the consequent behavior of the spectral response in the transition region, i.e., red-edge. This region, located between 680 and 750 nm, is highly sensitive to variations in the physiological state of the plants [56].

Radiometric sensors can be passive, if they use the solar radiation or other external light sources, or active, if they use their own artificial light source. Active sensors are less affected by variations in external lighting conditions and operate by emitting radiations in specific wavelength ranges and recording the reflected radiation from the plant or by emitting a polychromatic light and record the reflected radiation as well. Depending on the radiometric resolution, sensors can be classified into multispectral, if they have few broad- or narrow-bands (less than 20), or hyperspectral, if they have hundreds of narrow-bands.

Two commercial radiometric sensors are OptRx (Ag Leader Technology, Ames, IA, USA) (Figure 4), which is used to assess the vigor of the crop, and CropSpec (Topcon, Livermore, CA, USA), which measure plants reflectance to determine chlorophyll content, that is correlated to nitrogen concentration in leaves.

With respect to UAV, the proximal monitoring of vineyards vigor by radiometric sensors has the following drawbacks:

- the survey time is much higher as all the vineyards rows have to be passed through;
- the total monitoring cost is higher, due to the higher survey time;
- the data collected are tabular (GPS coordinates and corresponding NDVI values) so that they have to be converted in a map in the post-processing by interpolating the point values, which are usually collected every 30–50 cm along the vine rows and often every two rows;
- the final NDVI map is less precise as it results from an interpolation of sparse points collected along the rows.

On the other hand, the punctual data collected by radiometric sensors are more precise due to the higher resolution. In addition, another benefit of active radiometric sensors is the possibility to make the monitoring by night, as the vegetation is lighted up by the sensor itself.

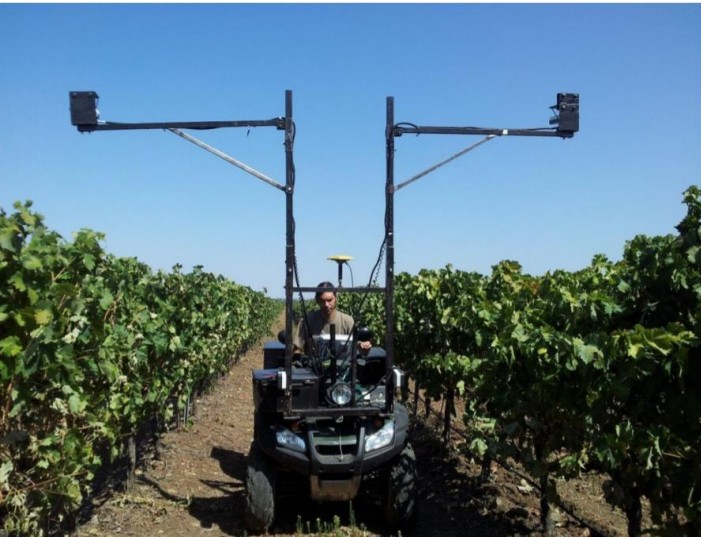

**Figure 4.** Proximal multispectral sensor (OptRx ACS-430, Ag Leader Technology, Ames, IA, USA) for NDVI mapping of vineyard canopy (reproduced from [57], with permission from *SOIL*, 2015).

Fluorometers

Another type of optical sensor is the fluorometer, which measures the fluorescence, a phenomenon that represents a form of dissipation of light energy by chlorophyll, that can be used to early assess the onset of a stress condition in the plant.

There are different types of fluorometer in the market like Hansatech FM2, Optisciences PSK Plant Stress Kit, Fluor Pen. These sensors operate with an emission of pulsed and modulated light. In practice, a typical fluorometer consists of a light emitter with a peak at 635 mm, a light emitter with a peak in the far red and a detector sensitive to radiation with a peak at 710–715 nm [9].

The variable derived from the fluorescence measurement that more than others represents the functional state of the plant, and therefore its photosynthetic potential, is the electronic transport rate (ETR). The linear combination of the ETR with other easily measurable variables (e.g., leaf and air temperatures) have allowed to set up an index ($I_{PL}$) capable of estimating the rate of net photosynthesis of the leaves [58].

VitiCanopy

VitiCanopy [59] (Figure 5) is one of the most used apps to measure the leaf area index (LAI) in vineyards through images acquired by a smartphone/tablet. Available for iOS operating systems, the app allows to calculate both the LAI and the porosity of the canopy in order to quickly and easily monitor the growth of the vines and the vigor of the vineyards.

The LAI, defined as the ratio between the total leaf surface and the surface of the soil on which the leaves are projected, is a very important measure in defining the vegetative-productive balance of the vineyard. The LAI can be measured with direct methods, which consist in destructive sampling of the leaves and their scanning, to obtain the total surface, or by weighting the samples to get the relationship between leaves area and leaves weight. These direct methods are normally quite accurate; however, they are destructive and very labor-intensive. Indirect methods are faster and based on photographic tools; however, they often require specialized personnel and they are quite expensive.

VitiCanopy uses the camera of mobile devices to obtain a photo of the canopy. The image is then analyzed to get the LAI, porosity and other descriptive parameters of the canopy architecture. Each result obtained with VitiCanopy is georeferenced so that it can be represented using a map, which allow to view the variation in the vigor of the entire vineyard. The app can be used by winemakers or researchers to monitor the spatial/temporal variability of the canopy growth and architecture in the vineyard.

These measures can be used for vineyard management (i.e., agronomic interventions), but also, they can be related to the quantity and quality of the final production.

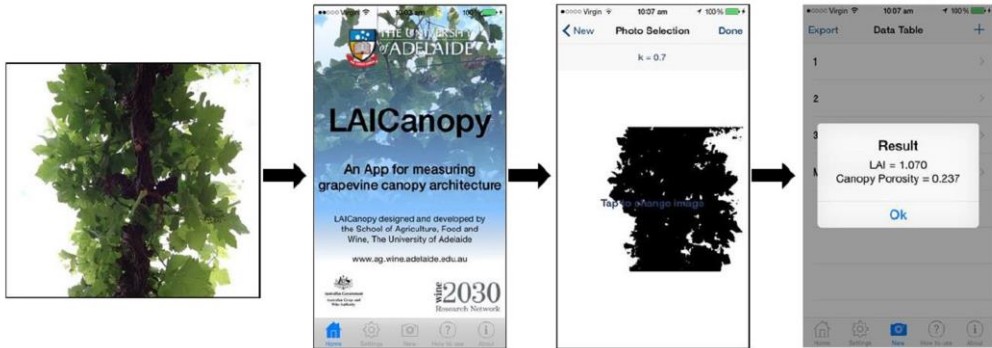

**Figure 5.** VitiCanopy app (Bruno Tisseyre, www.agrotic.org).

### 3.2.2. Soil

An important application of PV techniques is the proximal monitoring of soil variability, which includes the use of a wide range of sensors. The measurement of the electrical conductivity (EC) of the soil can be done by mobile platforms equipped with electromagnetic sensors and GPS [60,61]. It is strongly correlated with many soil properties, such as texture and depth, water retention capacity, organic matter content and salinity.

The sensors used for this type of measurements are electrical resistivity sensors (invasive) or electromagnetic induction sensors (non-invasive). The first type is used to check the resistivity (i.e., the inverse of the conductivity) of a given type of soil, generating electrical currents in the soil with an electrode and then measuring the potential difference thanks to a second receiver electrode. Among the commercial systems available, the Veris 3100 (Veris Technologies Inc, Salina, KS, USA) (Figure 6) and the Automatic Resistivity Profiling (ARP) system (Geocarta Ltd., Paris, France) are the most common. On the other hand, the principle of operation of electromagnetic induction sensors involves the generation of a magnetic field that induces an electric current in the soil, which in turn creates a second magnetic field proportional to its conductivity, which is measured by the sensor. Some devices on the market are DualEM (DualEM, Milton, ON, Canada) and EM-31 and EM-38 (Geonics Ltd., Mississauga, ON, Canada). There are also newly developed sensors to measure the pH, the nitrogen and potassium content and the infrared reflectance, as well as ground penetrating radar and radiometers.

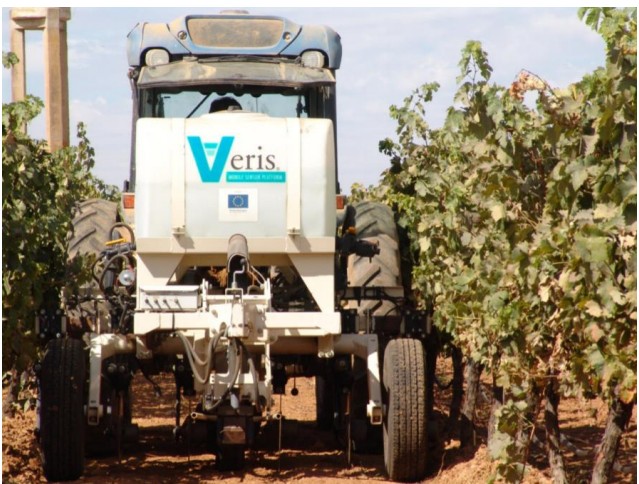

**Figure 6.** Mobile sensor platform Veris 3150 for ECa mapping (reproduced from [57], with permission from *SOIL*, 2015).

Soil properties play an important role in viticulture, since knowing the spatial variability of soil characteristics allows a better understanding of the variability of the physiological response of the vines [13]. The spatial variability of soil characteristics within a vineyard can be mapped at high detail by proximal sensing methods through the use of different type of sensors, like spectrometers, georesistivimeters and electromagnetic induction sensors [12]. This is particularly important in PV, since the homogeneous zone mapping within one or more vineyards allows differentiating the grapes on the basis of their potentiality and typicality, highlighting the "terroir effect" over the years [12]. In addition, during the planning of new vineyards, accurate information about soil features and their spatial variability are usually needed [62].

Geophysical Systems

The most commonly used sensors for the proximal detection of the characteristics of soils are the geophysical ones, which are based on the introduction of a current into the soil and the measurement of its potential drop, that is in turn directly related to the EC of the soil itself. Geophysical sensors allow to analyze the soil and characterize it in terms of EC, texture, organic matter content and moisture. In fact, each type of material has its own specific EC, so that from the measurement of this parameter the characteristics of the soil can be assessed. For example, the rocky substrate has EC values generally lower than 2–3 mS /m, sand has values between 1 and 10 mS /m, clay between 25 and 100 mS/m, while water can vary the EC depending on the dissolved salts from a few mS /m up to about 1000 mS /m [63].

The EC measured by geophysical sensors is related to the soil texture, the degree of humidity, the salinity, the porosity or the degree of compaction, the presence of gravel and pebbles. Depending on the type of sensor used, the EC of the first 40–50 cm of soil and up to a depth of a few meters can be measured. The data detected by the sensor will be an averaged EC of the volume of soil investigated, also called apparent electrical conductivity (ECa) [63].

Geophysical sensors can be invasive (mobile soil resistance-meters, Figure 7a), which measure the apparent resistivity of the soil by using direct contact electrodes, or non-invasive (electromagnetic induction sensors, ground penetrating radars), which allow to assess different soil properties (texture, water content, depth, porosity, etc.) by using electromagnetic fields. The electromagnetic induction sensor consists of two magnets, in which the primary emits a magnetic field, that generates an induced current in the soil, while the secondary one receives the magnetic field generated by these induced currents: the ECa of the soil is proportional to the ratio between the two magnetic fields.

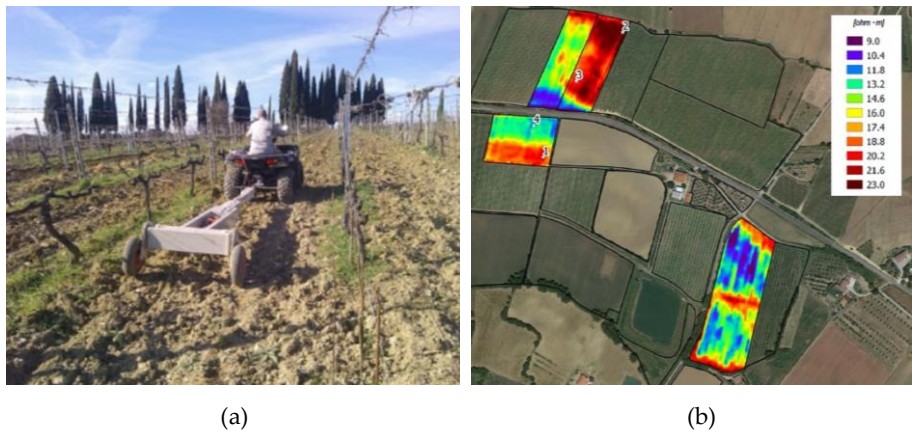

(a)　　　　　　　　　　　　　　　　　　　　　　　　　　　(b)

**Figure 7.** (**a**) Mobile soil resistance meters (reproduced from [63], with permission from *Georgofili*, 2017); (**b**) Electrical Resistivity map 0–50 cm (SO.IN.G. strutture & ambiente S.r.l., Cesa, Arezzo, Italy, 2019).

A typical electrical resistivity map is reported in Figure 7b.

Another type of non-invasive sensors are the ground penetrating radars, that allow to reconstruct the stratification of the soil through the emission of electromagnetic pulses and phenomena of reflection and refraction of the different materials of which the soil is made up.

Spectroradiometers

Spectroscopic techniques are based on the effects of the interaction between electromagnetic radiation and matter. The vis-nir spectrometry is based on the measurement of the spectral reflectance, i.e., the ratio between the radiation reflected from the analyzed surface and the one incident on it. Each soil has its own spectral signature which represents the intensity of the reflected radiation as a function of the wavelength and which depends on the intrinsic spectral behavior of heterogeneous combinations of minerals, organic matter and water.

Spectroradiometers are tools capable of detecting variations in mineralogy and chemistry of soils up to 50 cm of depth. These sensors can be gamma-ray or vis-nir reflectance based. The first type (i.e., gamma-ray spectroradiometers) consists of a scintillator crystal, generally made of cesium iodide (Csl) or sodium iodide (Nal), which emits photons when hit by gamma rays. The use of this tool is optimal when the main cause of soil variability is already known, for example, the type of substrate, surface stoniness or soil texture. Gamma-ray spectrometry is usually adopted to analyze large areas, in order to map the variability of soils linked to the first substrate, but also to scan small areas for detailed analysis of texture, surface porosity and carbonates [15].

Vis-nir reflectance spectroradiometers are gaining importance as they have advantages over other measurement methods. Firstly, this technique is rapid, relatively cheap, requires less sample preparation time and does not require the use of chemical reagents. Secondly, numerous soil properties can be quantified simultaneously, directly or indirectly, by a single scan. Finally, it can be used both in the laboratory, under controlled lighting conditions, or adapted to carry out measurements in the field, in stationary or mobile (continuous) conditions. With this tool it is possible to study the organic matter content, the water content, the soil texture and the mineralogical characteristics. Numerous researches have also concerned the quantification of the availability of nutrients, in particular macronutrients, pH, cation exchange capacity, structural properties, fractions of organic matter [15].

### 3.2.3. Grape Quality

Non-destructive monitoring of grape quality parameters is based on optical sensors designed as manual devices, i.e., tools carried by an operator, used for proximal georeferenced measurements.

Multiplex (Force-A, Orsay, France) [64] (Figure 8a) is a portable multiparametric fluorometer dedicated to the measurement of grapes and leaves parameters. By means of 4 excitation wavelengths and 3 sensing wavelengths, the Multiplex measures up to 9 fluorescence signals. In addition to these signals, Multiplex also assess plant physiological indices related to the content of chlorophyll, flavanols and anthocyanins in the leaves and grapes [65]. Multiplex was mainly introduced to estimate the nitrogen status of grapevine leaves. About this, on-field measurements have found reliable correlations between the nitrogen balance index (NBI) in leaves, the nitrogen in the must, wood pruning and leaves biomass [66].

Another sensor used to assess the quality of the grapes is the Spectron (Pellenc SA, Pertuis, France) (Figure 8b), a portable spectrophotometer with integrated GPS, designed to monitor grape ripeness through the non-destructive measurement of quality-related parameters, like sugar, acidity, anthocyanins and water content [9].

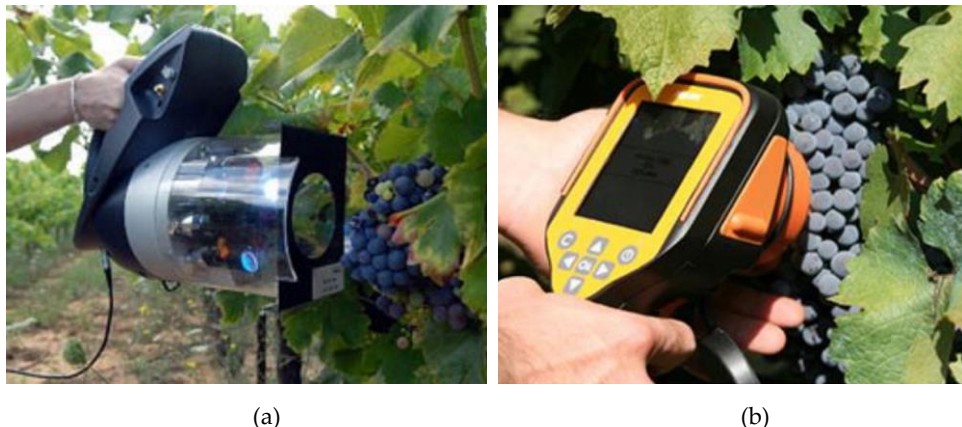

(a)　　　　　　　　　　　　　　　　　　　　(b)

**Figure 8.** Grape quality sensors (reproduced from [9], with permission from *International Journal of Wine Research*, 2015: (**a**) Multiplex (Force-A, Orsay, France); (**b**) Spectron (Pellenc SA, Pertuis, France).

## 4. Variable Rate Machines

Variable rate technology (VRT) machines allow to make more precise and automated field operations based on a prescription map expressed in doses of fertilizer, water, pesticide or defoliation rate. The harvesting operation can also be performed using a VRT machine with two hoppers that collect grapes of different quality.

Typical VRT machines for vineyard are reported in Figure 9.

The aim of carrying out VRT operations is to optimize the release of chemical inputs and to reduce the variability of the vineyard, trying to make it more homogeneous in terms of vigor [67]. Additionally, the application of VRT for harvesting can increase the wine quality by collecting grapes coming from different vigor zones in different hoppers [15].

The prescription maps are generated from georeferenced vigor maps that can be acquired by satellite, aircraft, UAV or with sensors mounted on tractors. Depending on the vigor class, the agronomist will express a corresponding dose which then the machine, by reading the map, will independently distribute. This process is possible thanks to the GPS module mounted on the VRT machine and its electronics, consisting of control units and proportional servo-valves.

Usually, in areas with low vigor there is a quantitatively lower grape production with a better quality and early ripening, while the opposite situation can be found in high vigor areas [68,69].

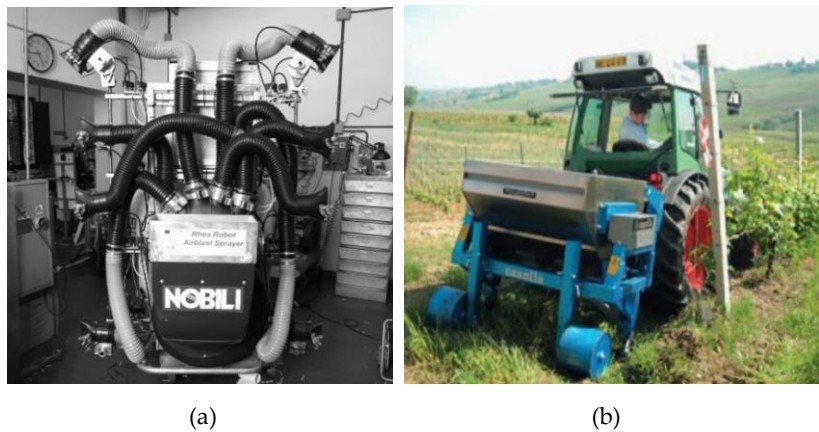

(a)　　　　　　　　　　　　　　　　　　　　(b)

**Figure 9.** VRT machines: (**a**) VRT sprayer [67] (reproduced from [67] with permission from *Ist Int. Workshop Vineyard Mech. Grape Wine Qual. Piacenza Acta Hortic*, 2013; (**b**) VRT fertilizer (reproduced from [69] with permission from *Universitat Politecnica de Valencia*, 2015).

The most interesting thematic maps for the winemakers (in addition to the vigor map) can be the ones related to yield, acidity, sugars, polyphenols and anthocyanin.

In order to have a standardized system of communication between tractors, software and various equipment and to allow the exchange of data and information with a universal language through a single control console integrated in the tractor cab, the ISOBUS or ISO 11783 protocol has been created [70]. This is the result of an agreement between the main agricultural machinery producers to solve compatibility problems, by standardizing and regulating the communication between different brands.

Thanks to ISOBUS, the cab becomes an authentic on-board computer, which allows to manage the machine and the equipment, guaranteeing the exchange of information.

In terms of economic savings, a variable rate fertilization can save up to 30% of product [71] and a variable rate irroration of pesticides can save up to 30% of product and increase the profit up to 20% [15].

## 5. Robotics

Robotics applied to agriculture will be the most important challenge of the next 10 years. In fact, these technologies will improve and automate agricultural processes, in view of the increase in world population and food needs, restrictions on pesticides and to increase eco-sustainability, in line with the European Green Deal and the "Farm-to-Fork" Strategy of the European Commission, and in the case of labor shortages, a very common problem in agriculture.

The use of robotics in PV is still at a prototype stage, but driverless agricultural robots (agbots or farmbots, Figure 10) are considered the future of agriculture [72].

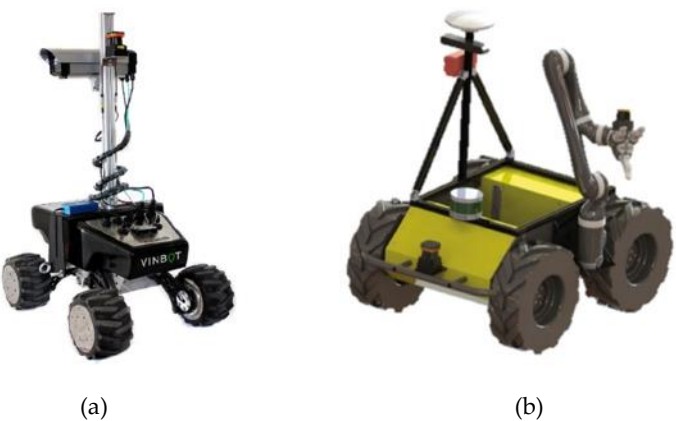

(a) (b)

**Figure 10.** Agbots: (**a**) Vinbot robot platform (reproduced from [73] with permission from *Jones, G.; Doran, N., Eds.*, 2016); (**b**) GRAPE (GroundRobot for vineyArd Monitoring and ProtEction) (reproduced from [74] with permission from *Springer*, 2017).

In recent years, there has been an important effort to find efficient innovative robotic technologies in agriculture, but the costs to implement these technologies are high: In the near future, it is foreseen an exponential increase of high-performance solutions at lower costs than the actual ones, which are between 50,000 € and 120,000 €, depending on the type of sensors mounted [9].

In general, agbots are usually equipped with RTK (real time kinematics) GPS, in order to follow the correct work-paths along the vine rows, computer vision sensors for obstacles detection and non-invasive sensing technologies like RGB, multispectral, fluorescence and thermal sensors. These systems are designed to perform a proximal monitoring of various parameters like yield, vigor, water stress and grapes quality, thus providing a decision support system to winegrowers for improving the vineyard management. Other systems are equipped with LiDAR sensors for 3D canopy reconstruction of the canopy, that can be useful to monitor the productivity [75,76].

In addition to these sensing agbots, some new developments are being done using artificial intelligence (AI) algorithms to guide the agbots in a series of more practical operations, like defoliation and selective grape harvesting by assessing the optimal ripening [9,75] and also the automatic pheromone distribution [74], spraying [77] and estimating pruning weights [78].

### 6. DSS and WSN

DSS (decision support systems) consist of at least five main components:

- a system for acquiring data relating to the cultivation environment, from multiple sources, which flow asynchronously to the DSS;
- a structure of interdependent databases that collects, organizes and performs a quality control of this data;
- sophisticated analysis algorithms (i.e., mathematical models) that allow the transition from raw data to processed data;
- automatic interpretation procedures that allow to pass from the processed data to the agronomic advice;
- a graphical interface that allows the user to access and interact with the DSS.

The acquired data usually come from multiple sources like weather stations (Figure 11) and wireless sensors placed in the field, up to image acquisition systems from fixed or mobile locations, both proximal (tractors) and remote (UAVs, aircraft, satellites), and also from IT tools (e.g., smartphones).

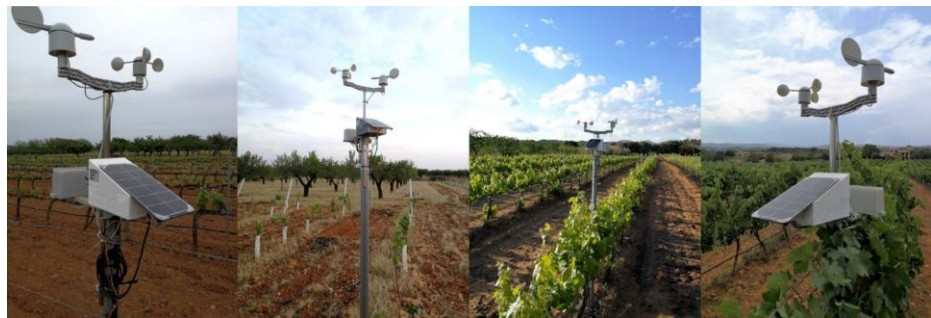

**Figure 11.** Weather stations to monitor the vineyard meteorological data (reproduced from [79] with permission from *Sensors*, 2020).

Wireless sensor network (WSN) technologies provide a useful and efficient tool for remote and real-time monitoring of important variables involved in grape production, processing the data and transmitting the required information to users. A WSN is a network of multiple wireless nodes connected together to monitor the physical parameters of the environment. The nodes are independent and installed in areas representative of the vineyard variability, also identifiable with the information provided by a vigor map [9]. Each wireless node is comprised of a radio transceiver, a micro-controller, sensors, an antenna and other circuitry that enables the communication to a gateway in order to transmit the information collected by the sensors [80]. The primary application of a WSN is the acquisition of micrometeorological parameters at canopy and soil levels. The sensors acquire these data and send the collected information to the controller, which further transmits this information to the cloud or to a portable device.

The most used WSN protocols communication in agriculture are [80]:

- Cellular (GSM, 3G, 4G), which is the most suitable for applications that require a very high frequency of data but it is also expensive and with high power consumption;
- 6LoWPAN, which is an IP-based communication protocol with low cost, low bandwidth and low power consumption;

- ZigBee, which is a wireless communication protocol with a flexible structure and a high battery life but it has a short operational range with low data speed and it is also less secure compared to Wi-Fi-based systems;
- BLE, which is a protocol similar to Bluetooth technology, has low bandwidth and short operational range (i.e., 10 m). The main advantages of this system are low setup time and low power consumption;
- Wi-Fi, which is the most common protocol that allows devices to communicate together thanks to a wireless network;
- LoRaWAN, which is a very common protocol used in agriculture due to its possibility to cover wide areas along with a low power consumption.

## 7. Conclusions

This review is intended to analyze the state of the art of PV technologies, whose main goals are the costs reduction in crop management by improving yield and quality of the production, ensuring traceability and environmental sustainability with a lower and/or more reasonable use of chemical inputs and water.

PV monitoring tools like UAVs, aircrafts, satellites and proximal sensors can be used together in order to have a full decision support system (DSS) which allow winegrowers and agronomists to take better decisions at the right time and place. The final result is an operational map (i.e., prescription map) that can be loaded into variable rate machines (VRT) to optimize field operations like fertilization, defoliation, irroration, phytosanitary treatments and harvesting.

Robotics is also being developed in a wide range of applications and it will be one of the most important agriculture challenges in the next 10 years.

In general, there are some barriers to the full adoption of PV technics, above all due to the implementation costs of these technologies, with unknown economic savings, and to the lack of technicians capable to properly use these technologies.

**Author Contributions:** Conceptualization, P.S. and R.P.; methodology, S.-P.K.; resources, S.-P.K. and M.A.; data curation, S.-P.K. and M.A.; writing—original draft preparation, S.-P.K.; writing—review and editing, S.-P.K. and M.A.; supervision, R.P.; project administration, P.S. and R.P.; funding acquisition, P.S. All authors have read and agreed to the published version of the manuscript.

**Funding:** This research was funded by the Italian Ministry of Agriculture grant number DM 36510, date 20 December 2018.

**Institutional Review Board Statement:** Not applicable.

**Informed Consent Statement:** Not applicable.

**Data Availability Statement:** No new data were created or analyzed in this study. Data sharing is not applicable to this article.

**Conflicts of Interest:** The authors declare no conflict of interest.

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
