# Peer review of "State of the Art of Monitoring Technologies and Data Processing for Precision Viticulture"

_agriculture, doi:10.3390/agriculture11030201_

Round 1
Reviewer 1 Report
The article in general is ok but I think a SWOT matrix would be needed to evaluate the suitability of these technologies when carrying them out. But no mention is made of drawbacks or potential problems when implementing them.
I believe that an economic study of the possible use of these techniques would be necessary
Which of all the techniques explained is better and why? Which technique is more economic or profitable in different scenarios?
Reviewer 2 Report
The paper is scientifically beneficial and interesting with new approaches. Some minor English revision is recommended.
For instant two same words in abstract.
The paper is well constructed easily readable. The literature is very extensive.
The resolution of some photos are too low (figure 4,5,6…).
I would appreciate to add some information concerning UAV path planning.
